# Daily MODIS Snow Cover Maps for the European Alps from 2002 onwards at 250 m Horizontal Resolution Along with a Nearly Cloud-Free Version

**Michael Matiu** *[ID], **Alexander Jacob**[ID] **and Claudia Notarnicola**[ID]

Institute for Earth Observation, Eurac Research, 39100 Bolzano/Bozen, Italy; alexander.jacob@eurac.edu (A.J.); claudia.notarnicola@eurac.edu (C.N.)

\* Correspondence: michael.matiu@eurac.edu

**Abstract:** Snow cover dynamics impact a whole range of systems in mountain regions, from society to economy to ecology; and they also affect downstream regions. Monitoring and analyzing snow cover dynamics has been facilitated with remote sensing products. Here, we present two high-resolution daily snow cover data sets for the entire European Alps covering the years 2002 to 2019, and with automatic updates. The first is based on moderate resolution imaging spectroradiometer (MODIS) and its implementation is specifically tailored to the complex terrain, exploiting the highest possible resolution available of 250 m. The second is a nearly cloud-free product derived from the first using temporal and spatial filters, which reduce average cloud cover from 41.9% to less than 0.1%. Validation has been performed using an extensive network of 312 ground stations, and for the cloud filtering also with cross-validation. Average overall accuracies were 93% for the initial and 91.5% for the cloud-filtered product using the ground stations; and 95.3% for the cross-validation of the cloud-filter. The data can be accessed online and via the R and python programming languages. Possible applications of the data include but are not limited to hydrology, cryosphere and climate.

**Dataset:** http://doi.org/10.5281/zenodo.3566703

**Dataset License:** CC-BY-4.0

**Keywords:** remote sensing; cryosphere; hydrology

---

## 1. Summary

Snow is a key environmental parameter in mountain regions. However, measurements using in-situ data are sparse and heterogenous because of the complex terrain, and additionally the altitudinal gradient is not appropriately represented since there are only few stations above 2000m altitude. Remote sensing offers an alternative which is not limited in spatial or altitudinal aspects but hampered by cloud cover. Of the currently available remote sensing snow cover datasets, MODIS (moderate resolution imaging spectroradiometer) offers the best trade-off between spatial resolution, temporal resolution, and temporal extent, since it is available at 250m horizontal resolution, with daily acquisitions, and since the years 2000 (Terra satellite) and 2002 (Terra and Aqua satellites).

This paper shall present two data sets. The first is a MODIS based snow cover product for the European Alps, whose algorithms have been developed until 2013 [1], and which is publicly available since 2018 (from now on called EURAC_SNOW); before the data was made available upon request. The second data set is a nearly cloud-free version, which has been derived from the first data set by using temporal and spatial filters (from now on called EURAC_SNOW_CLOUDREMOVAL). The



presented paper is a data descriptor, and its intent is to summarize the methods and post-2013 changes for EURAC_SNOW, to describe the methods used to create EURAC_SNOW_CLOUDREMOVAL, to provide some validation results, and to show how to access both data sets as a whole or in parts.

This is explicitly not a research paper, since nothing new has been developed. However, the novelty is that the data is publicly available. There is, to the best of our knowledge, no 250m MODIS based snow cover product such as EURAC_SNOW freely available for the European Alps. Regarding the cloud removal, while the methods used to create EURAC_SNOW_CLOUDREMOVAL have been published before in several studies, the data was seldomly made publicly available. Moreover, the code used in the cloud removal is also supplied, so that users can choose to do a different sequence of temporal or spatial filters.

The data can be used, for example, in the cryosphere sciences, but also for hydrological or climatological purposes. It has already been used in internal projects such as SAO (Sentinel Alpine Observatory, http://sao.eurac.edu/), in regionally funded projects such as MONALISA (http://www.monalisa-project.eu/) or CRYOMON-SciPro (http://www.eurac.edu/en/research/projects/Pages/projectdetail4240.aspx), in EU-funded projects such as ECOPOTENTIAL (https://www.ecopotential-project.eu/), openEO (https://openeo.org/), CliRSnow (http://www.eurac.eu/en/research/projects/Pages/projectdetail4488.aspx), and in various publications [2–9].

## 2. Data Description

Both snow cover datasets cover the European Alps and parts of its surroundings (see Figure 1 for extent; sometimes this is also referred to as the Greater Alpine Region). The maps are provided as discrete (snow, land, cloud, water, no data) integer valued raster maps in TIFF (tagged image file format) format (see Table 1 for detailed specifications).

**Table 1.** Characteristics of the two data sets.

| Characteristic | EURAC_SNOW | EURAC_SNOW_CLOUDREMOVAL |
|---|---|---|
| Name (coverage) | EURAC_SNOW_MODIS_ALPS_LAEA | EURAC_SNOW_CLOUDREMOVAL _MODIS_ ALPS_LAEA |
| Bands | SnowMap, SnowQuality [1] | SnowMap |
| Data type (SnowMap) | Integer (0 = no data; 1 = snow; 2 = land; 3 = cloud; 4 and 5 = water bodies) | Integer (0 = no data; 1 = snow; 2 = land; 3 = cloud; 5 = water bodies/no data) |
| Data format | TIFF [2] | same |
| Projection (proj4 string) | Lambert azimuthal equal-area (+init = epsg:3035 + proj = laea + lat_0 = 52 + lon_0 = 10 + x_0 = 4321000 + y_0 = 3210000 + ellps = GRS80 + towgs84 = 0,0,0,0,0,0,0 + units = m + no_defs) | same |
| Spatial coverage | xmin–xmax: 3847098–4982446 ymin–ymax: 2208289–2872448 (longlat~4.2–19.0 E, 43.0–48.6 N) | same |
| Spatial resolution | nominal: 250 m actual in projection: 232 m | same |
| Temporal coverage | 3 July 2002–09 June 2019 (as of 30 September 2019; updated automatically) | 3 July 2002–31 May 2019 |
| Temporal resolution | daily | same |

[1] SnowQuality is also integer (0–255), and gives the distance to the classification threshold (higher values mean more confident in classification). [2] Tagged image file format.

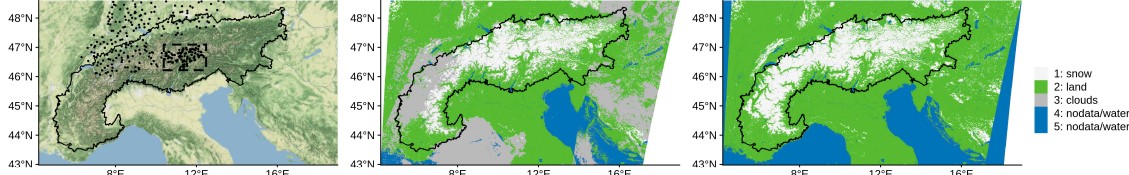

**Figure 1.** Map of the European Alps and example snow maps. Left: topographical map of the data extent; points indicate the location of the stations used in the validation; dashed rectangle marks the province of Bolzano (Italy). Middle: example snow map—MODIS based snow cover product for the European Alps (EURAC_SNOW) from Feb 28, 2019. Right: example of cloud-filtered map EURAC_SNOW_CLOUDREMOVAL from the same date.

The whole data as of publishing date can be accessed via Zenodo [10]. Access to the most up to date data is also possible online or via the programming languages R and python (see Table 2). The simplest form is online, but this is only recommended for a few single daily maps. For accessing the whole data set and being more flexible with spatial and temporal sub setting, it is recommended to use R or Python. Example scripts showing how to use the packages/modules are provided in the Supplementary Material.

The code that was used to generate EURAC_SNOW_CLOUDREMOVAL from EURAC_SNOW is available at https://gitlab.inf.unibz.it/earth_observation_public/modis_snow_cloud_removal. It is written in Python, and users can choose to redo the cloud removal or use different settings or filters based on their preferences or needs.

**Table 2.** Access options for the snow cover data. See Supplementary Material for example scripts for R and Python.

| Access | Platform | Package/Module Name | URL |
|---|---|---|---|
| WCPS [1] | online | | http://saocompute.eurac.edu/rasdaman/ows |
| | R | CubeR | https://gitlab.inf.unibz.it/earth_observation_public/CubeR |
| | python | modis_snow_cloudremoval | https://gitlab.inf.unibz.it/earth_observation_public/modis_snow_cloud_removal |
| openEO [2] | online | | http://editor.openeo.org/ |
| | R | openeo-r-client | https://github.com/Open-EO/openeo-r-client |
| | python | openeo-python-client | https://github.com/Open-EO/openeo-python-client |

[1] Web Coverage Processing Service. [2] For openEO the URL https://openeo.eurac.edu needs to be used.

## 3. Methods

### 3.1. EURAC_SNOW

#### 3.1.1. Summary of Algorithm Published in 2013

EURAC_SNOW is a snow cover product based on MODIS imagery and has been developed by Notarnicola et al. [1]. It aims at exploiting the 250 m bands in order to reach the highest resolution possible. This is relevant for mountain areas where the spatial heterogeneity cannot be captured completely from the original resolution (500 m) of MODIS snow products (MOD10A1/MYD10A1).

The main part of the algorithm relies on the detection of snow cover in open areas based on NDVI (normalized difference vegetation index) thresholds while the snow in forest is based on a multitemporal approach. In this multitemporal approach, the snow in forest was detected by comparing the actual image and a reference snow-free image acquired in the summer period. Being tailored for the European Alps, the algorithm considered adaptive thresholds for the main vegetation cover that is conifer, broadleaf and mixed forest. Moreover, the algorithm included a topographical correction of

the bands used in the snow detection in order to reduce the shadow effects which are typical during winter time.

Gridded fields allowed for a better analysis of the data; however, the quality of the information depended a lot on acquisition circumstances. For instance, off-nadir, the satellites observe areas were up to 10× larger than the pixel size [11]. This issue was to some extent alleviated by compositing Terra and Aqua and considering the viewing angle in selecting the "best" observation.

More detailed information on the algorithm is available in [1].

### 3.1.2. Post-2013 Changes

Since 2013, the workflow remained unchanged for the snow modules apart from adapting the thresholds to the new version 6 of MODIS products and to feedback received by the users who applied the snow cover maps in different contexts. This especially concerns the snow detection in forest which still remains an open issue in the optical remote sensing. The cloud module described in the 2013 version is no longer in use, instead the cloud information is taken directly from NASA's (National Aeronautics and Space Administration) MODIS product. Additionally, the quality layers from the 2013 version were dropped, and instead a different quality layer (band) was added, which is the distance to the classification threshold scaled to 0–255, where higher values mean more confidence in the classification.

### *3.2. EURAC_SNOW_CLOUDREMOVAL*

The usage of MODIS snow maps is hampered by clouds, which can obscure large parts of the data. For example, in Austria on average 63% of the region is covered by clouds, and even more in winter months, where snow observations are most needed [12]. Various techniques have been proposed to remove clouds in MODIS data. The simplest are based on combining information from Terra and Aqua, temporal filters using data from nearby dates [12], followed by spatial filters based on snow lines [13], which require a DEM (digital elevation model) as input, and combinations of spatial and temporal filters [14]. The more sophisticated methods, such as for example probabilistic merging of ground observations [15] or variational interpolation [16], however, require additional data, model training, or high performance computing.

As of 2019, NASA has planned to provide a daily cloud-free snow cover product at 500 m resolution [17], which is based on a backward temporal filter with a quality layer that identifies the number of days of the last cloud-free value used for filling [18]. In our case, there were no operational needs and no global datasets, so we were able to choose a sequence of cloud removal steps that go in both time directions and include some spatial information.

### 3.2.1. Proposed Filter Sequence for Cloud Removal

A cloud removal procedure was chosen that required minimal user input, and consisted of a sequence of temporal and spatial filters, which largely follows that applied in [14]. The first step consisted of preprocessing to alleviate the edge effects of snow and cloud misclassification. The second step involved a conservative temporal filter that was built upon the stationarity assumption that assumes that within a short cloudy period no change should happen. The third step was a spatial filter and this used the elevation dependence of snow cover. Finally, the fourth step was a greedy temporal filter that filled the remaining cloudy pixel with the next non-cloudy observation before or after. This last step was based on the assumption that snow cover is a gradual process and that sudden changes from snow to snow-free and vice versa are rare. An example of the sequence is provided for the province of Bolzano in Italy in Figure 2. In the following, these steps are explained in more detail.

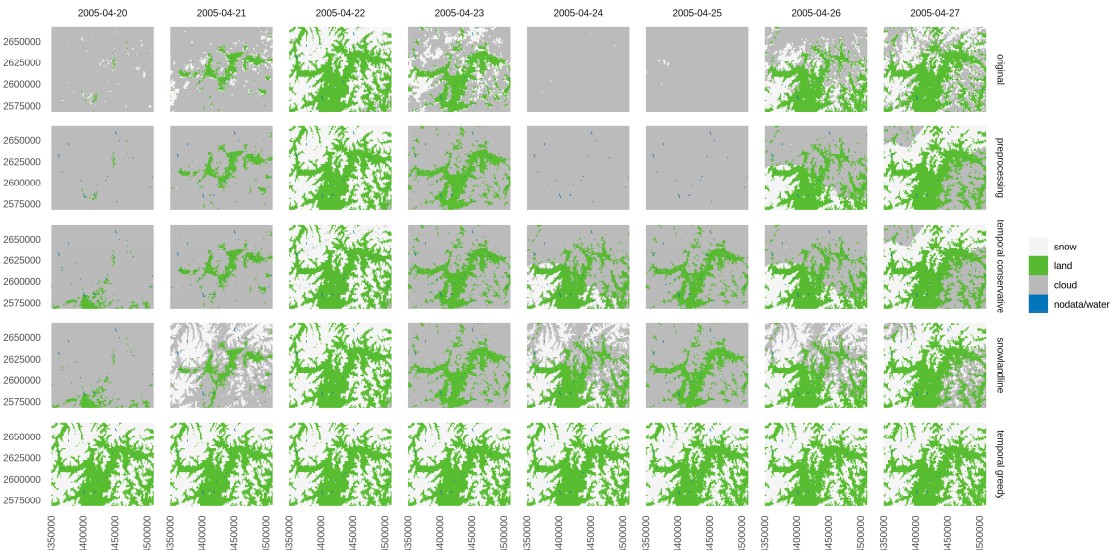

**Figure 2.** Example of the cloud removal sequence for the province of Bolzano in Italy (see Figure 1 for extent) for the end of April 2005. Map units (x and y axes) are projected coordinates. In the first row are the original images and in the other four the cloud removal steps detailed in the main text. Note that the procedure was applied to the entire images (e.g., for the calculation of snow and land lines), but the results shown here are just cropped to a smaller region for better illustration.

First, maps were preprocessed to remove edge and hole effects of snow and clouds. Snow and clouds are hard to distinguish, because of their similar optical reflectance. Since the cloud information was taken from the original NASA product, and only the remaining pixels were classified into snow and land, misclassifications can arise at cloud borders and within clouds, if the cloud information is inaccurate. For this, a simple mean filter was applied that reclassified only snow or cloud pixels based on their neighborhood. All other pixels (land, water, no data) were not used in this step. If the neighborhood contained more cloud than snow pixels, then the pixel was reclassified as clouds, otherwise if the neighborhood contained more snow than cloud pixels, it was reclassified as snow. A sufficiently large neighborhood (square of size 299 pixels centered on each pixel) was chosen to ensure that large scale cloud areas were adequately captured. This issue was most problematic in the non-winter season, so this step was only applied in the months April to October.

Second, a conservative temporal filter was applied using a time window of plus/minus two days. This classified a cloudy pixel as snow (land), if the previous and next days were both equally snow (land) at the very same pixel. If the previous and/or next day were also cloudy, then the values two days before and/or after were checked: if these were equally snow (land), then the pixel was made snow (land), otherwise it was left cloud.

Third, a spatial filter based on snow and land lines was applied, which required as additional input the altitude information of each pixel. The average altitude of all snow pixels at a given day gave the snow line. All cloudy pixels with an altitude above the snow line were then classified as snow. The procedure was applied analogously to the land line, and all cloudy pixels with altitude below the land line were then classified as land. This filter was not applied (a) if less than 50% of the image was cloud-free, (b) if the ratio of snow to land pixels was below 0.05, and (c) in the summer months (June–September). In these three cases, the snow line cannot be accurately determined (see also [14]).

Fourth, a greedy temporal filter was applied, which filled a cloudy pixel with the next snow or land observation at the same pixel. That means, the next non-cloudy observation, either forward or backward in time, was used to fill the cloudy pixel. If the next non-cloudy observation was an equal amount of days away both forward and backward, then the backward value was taken. Example: if a five-day cloud period was enclosed by first snow, and afterwards as land (S-CCCCC-L), then this step produced SSSLL as new values for the five cloud values. The maximum number of days to look for

non-cloudy observations was set to 10, since most observations were filled anyway in the first few days (see Figure S1). While in theory it would be possible to remove all clouds with such a procedure, using too large search windows might produce unrealistic results.

The whole sequence removed nearly all clouds (see Figures S2 and S3). The initial images had a mean cloud fraction of 41.9% (per-image cloud fraction almost uniformly distributed between 0 and 87.6%). After the first three steps, before the final greedy temporal filter, there was a mean cloud fraction of 22.0% but with a highly skewed distribution: a quarter of the images had less than 6.4% clouds, and half less than 15.6%. Finally, after the greedy temporal filter with 10 days maximum search radius, the per-image cloud fraction was less than 0.1% for 86.6% of the images, between 0.1–4% for 13.2% of the images, and up to 10% for the eight remaining images (0.01%).

### 3.2.2. Caveats of Filtering Sequence and Alternatives

Most importantly, all the filtering steps mentioned above inherit the errors in the original snow product, so removing clouds can only decreases the accuracy. However, each filtering step had a different effect on the accuracy (see also validation section below). For the temporal filters, accuracy was expected to decrease the farther away the next cloud-free observation was. If the cloud period was too long, then short snow or snow-free episodes were likely to be missed.

For the spatial filtering, the region used to determine the snow and land lines was the crucial component. If the region was too small, the snow and land lines might not be representative. If the region was too large, other factors such as shadowing effects and climatological aspects played important roles. Some of these effects were alleviated, because the mean altitude of snow and land pixels was used, and not the minimum or maximum, thus leaving some uncertainty altitude range in between that was not filled (see also [13]). Previous studies used this spatial filter for smaller region than presented here (e.g., approximately the NE quarter [13] or the SW quarter [14] seen in Figure 1), but also for larger regions (e.g., the whole contiguous US [16]). We decided to use the whole area to determine the snow and lines, because this step removed anyway only a small fraction of clouds compared to the other steps (see Table 3 in validation section) and there were only little differences if the snow and land lines were calculated for different regions and/or different aspects (see Table S1).

**Table 3.** Validation results for the snow maps and cloud removal steps using station data as ground truth.

| Cloud Removal Step | Mean Overall Accuracy | Mean Cloud Fraction | Station Data Used |
|---|---|---|---|
| Original [1] | 93.0% | 41.9% | 45.6% |
| Preprocessing | 94.0% | 40.0% | 45.0% |
| Temporal conservative | 94.6% | 25.4% | 60.6% |
| Snowlandline | 94.4% | 22.0% | 63.7% |
| Temporal greedy max10d [2] | 91.5% | 0.1% | 99.4% |

[1] Corresponds to data set EURAC_SNOW; [2] to EURAC_SNOW_CLOUDREMOVAL.

While we have proposed the above sequence, which, in our opinion, was a reasonable choice, we are aware that other users might think differently, and their needs might differ. Since the code is publicly available (https://gitlab.inf.unibz.it/earth_observation_public/modis_snow_cloud_removal) and built around the single filtering steps, users are encouraged to modify and adapt the cloud removal sequence to their needs.

### *3.3. Validation*

### 3.3.1. Summary of Initial EURAC_SNOW Validation from 2013

During the validation phase, EURAC_SNOW was compared with ground data and high resolution data such as Landsat imagery [19]. The comparison with ground data indicated an accuracy (ratio of

true positive and true negative to all) ranging from 82% to 94%. The lowest accuracies were mainly related to stations located in very steep slopes and along north facing slopes, which were strongly affected by shadowing effects found in very rugged terrain, and thus can lie in shadow in winter during the early morning acquisition. In the comparison with Landsat imagery, the average overall accuracy was around 88.1%. The mismatches may have been related to several factors such as the different acquisition time, misclassification related to cloud discrimination in the two sensors and the different sensor geometry. Outside the forested areas, the agreement reached 93.6%.

The EURAC snow cover area (SCA) products were also compared with original MODIS products (MOD10A1/MYD10A1). The accuracy ranged from 78% and 98.3% with an average of around 84% and where the biggest discrepancies were found in forested areas. Excluding forested areas, the average accuracy was around 90.2%.

### 3.3.2. New, Extended Validation of EURAC_SNOW and EURAC_SNOW_CLOUDREMOVAL with Station Data

The validation from 2013 with ground data covered large parts of the alps but was limited to the 2005/2006 season. In contrast, here we present further validation results for a different domain and for the whole time period based on a different network of ground stations (see Figure 1 for their location).

Snow depth data was acquired for 88 stations in the province of Bolzano in Northern Italy from the Hydrological Office of Bolzano, Italy, for 53 stations in Germany from the German Weather Service (DWD), and for 171 stations in Switzerland from the Federal Office of Meteorology and Climatology MeteoSwiss. During the common period with the remote sensing snow maps (July 2002–May 2019), there were in total more than 1.2 million of non-missing observations, yielding an average of 4037 observations per station (min 361, max 6176). Stations are known to be heterogeneously distributed in alpine terrain, concerning their location as well as the altitude. The same was true here: the frequency distribution of the station altitude and the underlying MODIS altitude differed (see Figure S4), however, the stations sampled large parts of the altitudinal distribution (until ~3000 m).

Snow depth was measured in cm, and was converted to a binary variable snow/land using a threshold of 5cm, which is widely used to calculate snow cover days, see e.g., [20]. Selecting the best threshold for the binary classification of snow versus snow-free was not a trivial issue. It depended on the region of interest, the altitude, as well as the season. We compared different thresholds of 1, 2, 5 or 10cm (see Table S2), and while there were minor differences between accuracies, the differences in accuracy between the cloud removal steps were independent of the threshold. Thus, for our purposes, we decided to use a single threshold of 5 cm.

In order to compare the point observation (station) to the gridded field (snow maps), we extracted the values for each pixel which contained a station. We also tested extracting $3 \times 3$ or $5 \times 5$ pixels centered on the station location but found that results were qualitatively similar to only using one pixel. Then we calculated overall accuracy (OA) as the sum of true positive (both station and remote sensing are snow) and true negative (both land) divided by the total number of cases (true positive + true negative + false positive + false negative; where positive = snow and negative = land). OA was calculated separately for each station. The total OA was then the weighted mean of station OA's with the number of observations as weights.

Table 3 shows the OA's for the different cloud removal steps averaged over the 312 stations along with the average cloud fraction of the whole domain and the fraction of station data used. Mean OA was 93.0% for the original images (data: EURAC_SNOW) with 41.9% of clouds. The preprocessing increased accuracy to 94.0% by reducing the number of false snow classifications outside of winter. The conservative temporal filtering also increased OA further to 94.6%, most likely because there was more data to compare and the filtering itself introduces almost no error by definition. Finally, the 10-day greedy filter (data: EURAC_SNOW_CLOUDREMOVAL) reduced OA to 91.5%, which was only slightly less than the original images, with an average cloud fraction of less than 0.1%.

While these were average OAs, the OA differed substantially by month (Figure S5a). OAs were lowest in winter (January: 85.1% for the original and 83.3% for the cloud-filtered maps), average in spring (April: 94.9% for the original and 90.9% for the cloud-filtered maps), and highest in summer (July: 96.6% for the original and 97.7% for the cloud-filtered maps). We also evaluated the OA by year (Figure S5b) and found some variability (standard deviation of 0.7% for the original and 0.8% for the cloud-filtered maps), but no obvious systematic patterns.

The validation results were thus on the higher end side of the initial validation from 2013 [19], but well in the range of similar studies: 95.7% for the Italian part of the Western Alps [14], 90–96% for northeastern Afghanistan [21], 92.1% for Austria [12], 91% for Nepal [22], and 95.5% for the contiguous US [16].

### 3.3.3. Spatial Cross Validation of EURAC_SNOW_CLOUDREMOVAL

Station data offer a validation opportunity for remotely sensed snow and can be used to "see beneath" clouds. However, comparing points to grids suffers from scale issues and stations do not cover the continuous domain as well as the whole altitudinal gradient. Another option was to use other remote sensing products for evaluation. In the case of EURAC_SNOW, this has been already performed using Landsat imagery, and we did not want to repeat it here. Using Landsat or other satellites for EURAC_SNOW_CLOUDREMOVAL is difficult, because it is very unlikely to find days, where MODIS is cloudy (which is then filtered away) and Landsat (or another sensor) image is cloud-free—even with different acquisition times. Moreover, comparing products with different resolutions induces another sort of scale issue compared to point observations, and it is not possible to distinguish the influence of the cloud filtering steps.

Instead we adopted a cross-validation technique similar to what [14] performed, except in a more systematic way. The cross-validation consisted of randomly selecting one hydrological year (to reduce computation time), which was October 2013 to September 2014. Then, for each day in this period, the map was taken aside, a copy was made all clouds, and the cloud filtering sequence was applied to this deliberately missing copy using the non-missing neighboring dates. Consequently, the filtered map could be compared to the observed one, which was set aside, and was not used in the filtering. This should give an indication of the performance of the cloud filtering sequence.

The average accuracies were remarkably high, with 97.6% for the conservative temporal filter and the snow-land-line filter, and 95.3% for the 10-day greedy temporal filter. As with the station data, accuracies displayed a seasonal pattern (Figure S6). The average OA in the extended winter season (November–April) was 95.4% for the conservative temporal filter and the snow-land-line filter, and 91.2% for the 10-day greedy temporal filter. Figure S7 shows example maps of two dates that sampled the average and worst OA in winter. In both cases, the cloud-filtering was able to delineate the snow distribution well in higher altitudes, however, for lower altitudes with short snow episodes the accuracy depended on how far the next non-cloudy observation was.

## 4. User Notes

### 4.1. Data Access

See Supplementary Material for example scripts in R and python on how to access temporal or spatial slices of the data via WCPS (Web Coverage Processing Service). Since openEO is under heavy development, no examples are provided, but users are referred to the respective github sites for up to date examples.

### 4.2. Example Usage

The cloud-filtered data EURAC_SNOW_CLOUDREMOVAL can be used, for example, to calculate the snow-covered area (SCA). Figure 3 shows the SCA in January for the four climatological regions of the Greater Alpine Region as defined in the HISTALP project (http://www.zamg.ac.at/histalp/project/

maps/gar_reg.php). Even from these only 17 years, a small decrease in SCA could be observed, which, however, differed by region.

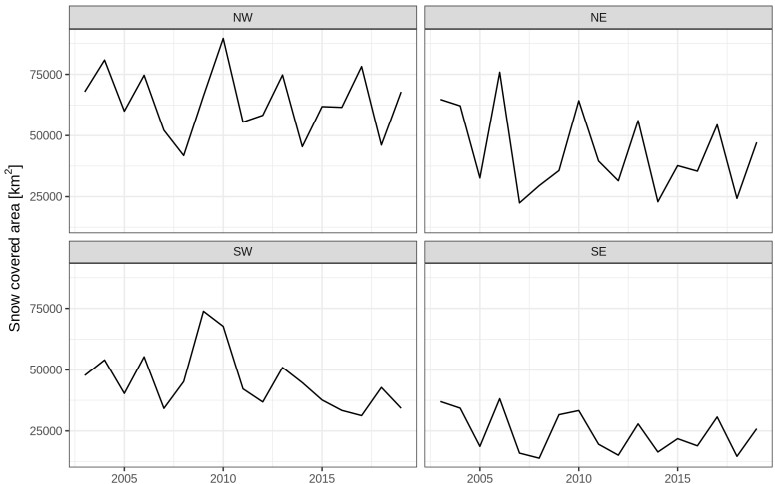

**Figure 3.** Average snow-covered area in January from the cloud-filtered data (EURAC_SNOW_CLOUDREMOVAL) for four climatological regions. The regions originate from the HISTALP project (see http://www.zamg.ac.at/histalp/project/maps/gar_reg.php) and are climatologically defined regions in the Greater Alpine Region (NW: north-west, NE: north-east, etc.).

*4.3. Other Cautionary Notes*

The data presented here is observational data, and with any observations, there is some inherent uncertainty. The snow cover products should not be taken as perfect, and their accuracies should be kept in mind for their usage.

The data can be used well for large areas; however, single pixel time series should be treated with caution. The accuracy of single pixels can be influenced by many variable factors, such as acquisition angle, timing of the season, shadowing effects of surrounding mountains, and land cover. Especially for forest covered areas (of evergreen coniferous), lowest accuracies were found. When averaging over large areas or longer periods, these effects cancel out to some extent.

The purpose of the cloud removal is to have an almost complete time series, which makes many analyses easier. However, the accuracy is lower than for the original product, especially in the transition seasons at the start and end of the snow season. The last temporal filter was designed to provide reasonable estimates when averaged over longer periods (weeks to months). Consequently, the daily values should be used very carefully, and especially those in spring.

**Supplementary Materials:** The following are available online at http://www.mdpi.com/2306-5729/5/1/1/s1, Figure S1: influence of the number of days to search in the greedy temporal filter, Figure S2: overview of the average monthly cloud fraction, Figure S3: cloud fraction distribution, Figure S4: frequency distribution of station and MODIS altitudes, Figure S5: overall accuracies of cloud removal by month and year, Figure S6: overall accuracies for cross-validation of the cloud removal, Figure S7: example maps for the cross-validation of the cloud filtering, Table S1: mean overall accuracies of the snowlandline filter for different regions, Table S2: influence of snow depth threshold on the overall accuracy, HTML (code examples).

**Author Contributions:** Conceptualization M.M. and C.N.; methodology M.M., A.J. and C.N; software M.M., A.J.; validation M.M.; data curation A.J.; writing—original draft preparation M.M.; writing—review and editing M.M., A.J. and C.N.; visualization M.M.; supervision C.N.; funding acquisition M.M. and C.N. All authors have read and agreed to the published version of the manuscript.

**Funding:** This project has received funding from the European Union's Horizon 2020 Research and Innovation Programme under the Marie Sklodowska-Curie grant agreement No 795310. The APC (article processing charge) was funded by the European Union's Horizon 2020 Research and Innovation Programme under the Marie Sklodowska-Curie grant agreement No 795310.

**Acknowledgments:** We acknowledge the Hydrological Office of Bolzano, Italy for the ground data. We also want to thank Valentina Premier, Carlo Marin and Mattia Callegari for discussions and the initial code pieces.

**Conflicts of Interest:** The authors declare no conflicts of interest. The funders had no role in the design of the study; in the collection, analyses, or interpretation of data; in the writing of the manuscript, or in the decision to publish the results.

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
