# Peer review of "Daily MODIS Snow Cover Maps for the European Alps from 2002 onwards at 250 m Horizontal Resolution Along with a Nearly Cloud-Free Version"

_data, 2002_

Round 1
Reviewer 1 Report
The manuscript by Matiu et al. developed a 250-m MODIS snow dataset. I find the presentation of the paper and the description of the methods are not always clear and some times confusing. Besides, in my opinion, this paper did not develop some new algorithms and methods. Thus, this study lacks novelty at this point and the value of the paper is weakened. I do not recommend a publication of this paper in Data at its current form. I give some detailed comments below.
1.L151-159. In my opinion, the spatial filter of the snow line is not fine enough or is problematic. The accuracy may be largely dependent on the window size. For example, the snow lines derived from a window size of 5km and 500km are apparently different. When the window size increases, the distribution of snow is not fully driven by elevation, but also other factors, e.g. rain-shadow effect, atmospheric circulation, vegetation. How did you know what window size fits best with your study region?
2. L160-165. The description of this step is confusing. Why 5-day sequence of SCCCCL is changed to a 7-day sequence of SSSSLLL?
3. L178-181. I did not get what steps were used by the workflow of Notarnicola (2013), and what steps are newly used here. Please make it clear.
Notarnicola, C.; Duguay, M.; Moelg, N.; Schellenberger, T.; Tetzlaff, A.; Monsorno, R.; Costa, A.; Steurer, C.; Zebisch, M. Snow Cover Maps from MODIS Images at 250 m Resolution, Part 1: Algorithm Description. Remote Sens. 2013, 5, 110–126.
4. L188. At different regions and elevations, the best threshold for classifying snow/land may be different. Have you tested the other thresholds? What is the sensitivity of accuracy to threshold changes?
5. Table 3 and Fig 1. Again, what are the differences of the algorithms used here and Notarnicola (2013)? Even some of these steps are newly used here compared to Notarnicola (2013), these methods have already been used by many researchers (e.g. Parajka et al., 2008, 2010; Gafurov et al, 2009). What is the novelty of this paper?
Parajka, J.; Blöschl, G. Spatio-temporal combination of MODIS images – potential for snow cover mapping. Water Resour. Res. 2008,
Parajka, J.; Pepe, M.; Rampini, A.; Rossi, S.; Blöschl, G. A regional snow-line method for estimating snow cover from MODIS during cloud cover. J. Hydrol. 2010, 381, 203–212.
Gafurov, A.; Bárdossy, A. Cloud removal methodology from MODIS snow cover product. Hydrol. Earth Syst. Sci. 2009, 13, 1361–1373.
Reviewer 2 Report
In this manuscript the authors present a daily nearly cloud-free snow cover dataset at ~250m resolution over the European Alps. The dataset was created by removing the cloud pixels in the previously published Eurac MODIS snow cover based on a series of filters. The resulting snow cover data contains less misclassification of snow based on the comparison with snow station measurements.
While the de-cloud snow data potentially can benefit the snow and hydrologic research, the manuscript does not have convincingly enough details on the data producing method and data validation; major concerns about the rationale of the methods and validating the data arose when reading these sections. I suggest the authors consider the following comments when revising the manuscript.
1. Several important related literatures are forgotten. For example Dozier et al, 2008 mentioned the fact that off nadir views from the satellite observe an area up to 10x the pixel size increasing error. How could the authors deal with this input data issue? The manuscript also ignores Dorothy Hall et al, 2010 which provides a recent gap filling method.
2. More details about the last filtering is needed. Why did the authors decide to choose the greedy temporal filter? Is it based on some of the characteristic of snow? Is there any hypothesis for this approach?. Based on the current manuscript, it is difficult find more details to help better understand the solution.
3. Regarding the data validation:
a) Use the point-scale measurement to validate the snow cover data cannot be convincing. While the snow recorded at each station can tell whether the snow cover data grid-cell misclassify snow or not, an important characteristics of the derived data is its delineation of the snow covered extent. If the de-cloud snowpack boundary happen to fall in between two observations, how can the two point-scale observations tell where the true snowpack boundary should be? I suggest comparing the cloud-removed snow cover data with some other spatially continuous snow cover data, e.g. the selected Landsat images that are not cloud contaminated. Landsat data should also cover the current validation period.
b) The validation is only conducted for the Bolzano province. More detail about the province is needed (e.g. terrain, climate information, elevation range). Is the region representative enough for the European Alps. To ensure the accuracy of entire dataset so to increase its utility, it is important to validate all the domain.
c) How did the cloud removal perform in dry years versus wet years?
4. Some examples of usage of the dataset would be very useful. Intuitive questions like how the snow extent changed during the past two decades, which areas of the Alps receive more snow can be answered by including more ‘result’ figures in the manuscript.
Round 2
Reviewer 1 Report
The manuscript by Matiu et al. tiled “Daily MODIS snow cover maps for the European Alps from 2002 onwards at 250m horizontal resolution along with a nearly cloud-free version” (Manuscript ID: data-648402) has been improved after the major revision. The authors have paid much efforts to revise this paper. Most of my previous concerns have been adequately addressed. Overall, this is a well-organized, clearly written manuscript that provides a new MODIS snow cover dataset for the Alps at 250m resolution, in which most of the cloud contamination has been removed, although all the processing steps were proposed by previous researchers. This dataset would be useful for the future research about the snow hydrology, climate change, water resources in the Alps. In terms of remaining revisions, minor proofreading would be recommended throughout the manuscript for small typos/errors.
Reviewer 2 Report
I appreciate the authors effort to address the comments. Since they have stated that the purpose of this article is to "only combined already available data and methods to produce data", I don't have an urge to ask more about its novelty.